# Dormancy Breaking of *Teramnus labialis* (L.f.) Spreng Seeds Is Affected by the Extent of Liquid Nitrogen Exposure

Yanier Acosta Fernández [1,*], Lianny Pérez Gómez [1], Dayami Fontes Marrero [2] and Marcos Edel Martinez Montero [2,*]

1 Laboratory for Plant Breeding and Conservation of Genetic Resources, Bioplantas Centre, University of Ciego de Ávila Máximo Gómez Báez, Modesto Reyes, Ciego de Ávila 65200, Cuba
2 Faculty of Agricultural Sciences, University of Ciego de Ávila Máximo Gómez Báez, Modesto Reyes, Ciego de Ávila 65200, Cuba
* Correspondence: yacfdez@gmail.com (Y.A.F.); cubaplantas@gmail.com (M.E.M.M.)

**Abstract:** *Teramnus labialis* (L.f.) Spreng shows dormancy as a result of impermeability of the seed coat, which requires scarification treatment before sowing. Liquid nitrogen (LN) as a scarifying treatment has recently been used on this species, with excellent results. However, moisture content and immersion time on LN are factors that may affect seed germination and dormancy break. This report studies (i) the effects of dehydration on *T. labialis* seed viability and germination and (ii) the appropriate moisture content and extent of LN to make this scarification an effective treatment. Moisture contents of 4%, 6%, 8%, and 10% fresh weight basis (FWB) and extension to LN for 15, 30, 45, and 60 min were performed. Seed viability did not change after seed dehydration up to a moisture content of 4% FWB, whereas the percentage of germination decreased as a result of increasing the percentage of hard seeds. The seed moisture content did not affect germination after immersion in LN, but at least 30 min of exposure was required for dormancy break. The mean germination time, germination index, and time to 50% germination improved with the increasing germination from 33% to 91% and a greater duration of immersion in LN. The dormancy of *T. labialis* seeds with a moisture content between 4% and 10% FWB was only broken when exposed to LN for 30 to 60 min.

**Keywords:** dehydration; germination; moisture content; scarification; seed coat



## 1. Introduction

Legumes (Fabaceae or Leguminosae) are one of the most important and widespread families worldwide [1]. There are numerous species in different genera (*Arachis*, *Centrosema*, *Clitoria*, *Desmodium*, *Devaux*, *Macroptilium*, *Neonotonia*, *Stylosanthes*, and *Teramnus*) of this family in tropical America [2,3]. *Teramnus labialis* (L.f.) Spreng belongs to the genus *Teramnus* and is a tropical herbaceous legume of great importance for agriculture and livestock [4–7].

However, establishing this species is affected by the low percentage of germinating seeds [6–8]. Physical dormancy (PY) in seeds, due to the impermeability of the seed coat to water and air, has been described as one of the main causes of low germination [9,10]. Despite the ecological advantages that PY bring, this mechanism is an obstacle to germination, emergence, and establishment of the plant [11,12]. Therefore, it is necessary to treat the seeds before sowing to achieve successful germination.

To overcome PY, it is important to eliminate seed impermeability by scarification treatment [11,13]. These treatments are divided into three different methods: mechanical, chemical, and physical [14–16]. In *T. labialis* seeds, sulfuric acid (chemical method) [17] and hot water (physical method) [18–20] have been used as scarification methods.

In recent years, the use of liquid nitrogen (−196 °C, LN) has been added as an effective treatment (physical method) to dormancy break and improve germination [9,10,21] and establishment in various plant species [21–23]. However, the effectiveness of scarification

treatment with LN depends on seed size, seed coat thickness, chemical composition [24], seed moisture content [25,26], and time of immersion in LN [27,28].

The moisture content can affect seed germination because intracellular water crystallizes during treatment with LN, which is lethal to embryonic tissue [29,30]. In addition, the extension to LN determines the efficacy of this scarification treatment, because the seeds must reach a cryogenic temperature when immersed in the scarifying agent [27,31,32]. Using scanning electron microscopy, for the first time, our group demonstrated that *T. labialis* species seeds with a moisture content of 9.8% fresh weight basis (FWB) showed the formation of multiple cracks in the seed coat when immersed in LN for 24 h, corroborating the findings of Acosta et al. [9]. In addition, using light microscopy, we found that seeds with a moisture content of 7.98% FWB at harvest that were immersed in LN for 24 h had an open hilar region and the seed coat became permeable to water, compromising its integrity [10]. These criteria served as the basis for conducting this research, aiming to (1) determine the effect of seed dehydration on the viability and germination of *T. labialis* seeds and (2) determine the moisture content and appropriate times of immersion in LN to make this scarification an effective treatment.

## 2. Materials and Methods

### 2.1. Seed Material

Fifty plants of *T. labialis* (L.f.) Spreng were grown from July 2017 to March 2018 in the city of Ciego de Ávila, Cuba (21°89′14.07″ N, 78°69′67.53″ W). Seeds were harvested on 12 March 2018, when 80% of the pods were ripe.

### 2.2. Seed Physiological Evaluations

Seed moisture content (SMC): The oven-drying method was used to determine the SMC [33]. Three samples of 50 seeds each were collected, placed in a porcelain container, and weighed on an analytical balance (SARTORIUS, BL 1500). The containers were placed in an oven (HS62A) at a temperature of 130 °C until the weight was constant. The containers were then placed in a desiccator containing silica gel and allowed to cool for 45 min. The samples were then reweighed, and the SMC was calculated based on the fresh weight using the following formula (1):

$$SMC = \frac{\text{fresh weight} - \text{dry weight}}{\text{fresh weight}}$$

where SMC is the moisture content of seeds, fresh weight is the weight of seeds before placing in the oven, and dry weight is the weight of seeds after drying in the oven.

Seed viability: Three replicates of 50 seeds were collected to determine viability using the topographic tetrazolium assay [34]. A small incision was made in the coat of all seeds with a scalpel in the region opposite the thread to facilitate imbibition. Each sample was placed in a Petri dish containing felt paper previously moistened with 10 mL of distilled water for 24 h. The coat of each seed was then removed to expose the embryo. The embryos were again placed in the Petri dishes, but this time 10 mL of a 1% solution of 2, 3, 5 triphenyl-2H-tetrazolium chloride (TTC) was added. After 6 h in the dark, embryos were categorized by their coloration as (1) viable if they were completely colored deep red, pale red, or with discolored sections, and (2) non-viable if they retained their original color; the results were expressed as the percentage of viable embryos to the total number.

Seed germination: Four replicates of 25 seeds were tested for germination [10]. Plates containing the seeds were moved into a growth chamber (TOP Cloudagri, RTOP-1000 B/D, China) set at a constant temperature of $30 \pm 1\ °C$ under a 16 h light/8 h dark photoperiod with a photosynthetic photon flux density of 80 µmol·m$^{-2}$·s$^{-1}$ and 80% relative humidity. Seeds were placed on a single sheet of filter paper in 90 mm diameter plastic Petri dishes (unsealed). The filter paper was moistened with 5 mL of distilled water every 7 days. The number of germinated seeds was analyzed after 28 days in a growth chamber. A seed was considered germinated when its radicle was ≥2 mm long. The results were expressed as the percentage of germinated seeds to the total number of initial seeds [35].

### 2.3. Seed Drying Experiments

Seeds were dehydrated to different moisture contents (10%, 8%, 6%, and 4% FWB) before scarification. Dehydration was carried out according to the method proposed by Kameswara et al. [33] with a ratio of blue self-indicating silica gel to seeds of 3:1 (weight of blue self-indicating silica gel to the weight of seeds). The time required for the seeds to reach each moisture content was estimated from a standard dehydration curve previously established for these seeds (Figure 1).

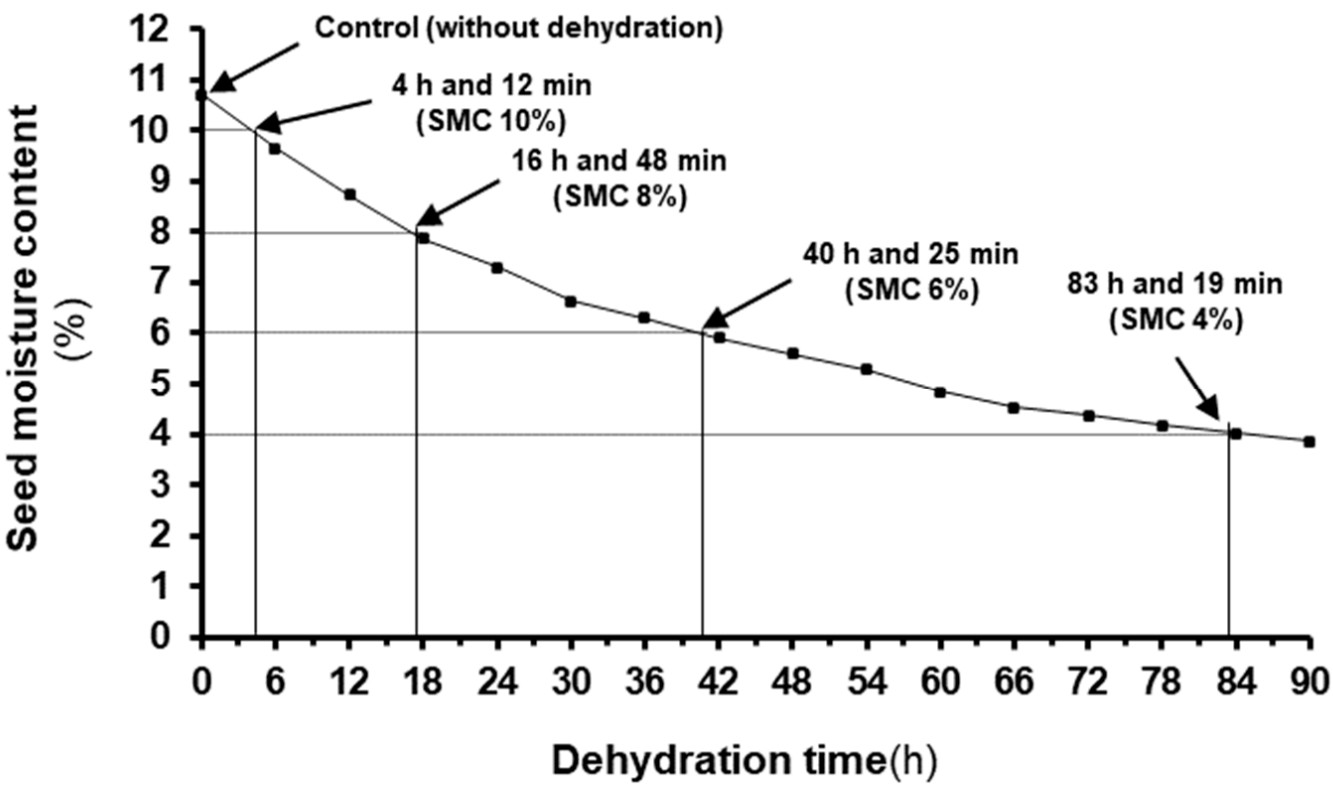

**Figure 1.** Standard curve for the dehydration of *Teramnus labialis* (L.f.) Spreng seeds. The graph shows the dehydration times required for the seeds to reach the target seed moisture content (SMC).

In total, 1250 seeds were selected. Three replicates of 50 seeds, i.e., a total of 150 seeds per treatment for each moisture content, including controls without dehydration, were evaluated for viability (%). At germination, four replicates of 25 seeds, i.e., a total of 100 seeds per treatment for each moisture content, were analyzed. Therefore, 750 seeds were used for viability (%) and 500 seeds were used for germination (%) to determine whether these physio-logical indicators were affected by dehydration. In addition, hard seeds (ungerminated seeds that were obviously viable) were also calculated [36].

*2.4. LN Scarification Experiments*

Dehydrated seeds with different moisture contents (10%, 8%, 6%, and 4% FWB) were placed in polypropylene cryovials and immersed directly in LN for different times (15, 30, 45, and 60 min). For rewarming, seeds were removed from the cryovial and kept at room temperature (25 °C) for 2 h [10].

For the 16th treatment combination, 1600 seeds in total were used (with four replicates of 25 seeds, i.e., a total of 100 seeds per treatment for four moisture contents and 100 seeds for four immersion cycles in LN). The percentages of germinated seeds (radicle ≥2 mm long), hard seeds (ungerminated seeds that were obviously viable), and dead seeds (seeds with fungal infection and obviously not viable) were determined. The germination times of 50% of the seeds (T50, days), mean germination times (MGTs, days), and germination index (GI, seeds days$^{-1}$) were also calculated [35].

*2.5. Histological Evaluation of Seed Coat*

Twenty seeds with a moisture content of 8% FWB from each exposure time to LN (15, 30, 45, and 60 min) and the control treatment (without exposure to LN) were randomly selected for histological and anatomical studies. Seeds were fixed in formalin/acetic acid/alcohol, dehydrated in an ethanol series, and embedded in paraffin, according to Johansen [37]. Then, the seeds were cut into transverse sections (5 μm thick) using a KD-97 202A hand-held rotary microtome. They were then collected on microscopic slides coated with a 1% (*v:v*) gelatin solution. The sections were stained with 1% safranin for 24 h and then viewed with a Zeiss® microscope connected to a Canon® Power Shot A 630 digital camera (200x magnification).

*2.6. Statistical Analysis*

All data were statistically analyzed using SPSS (version 8.0 for Windows, SPSS Inc., New York, NY, USA). All data were tested for normality with a Shapiro–Wilk test. Mean values of viable, germinated, and hard seeds were compared using one-way ANOVA and Tukey's test at $p \leq 0.05$. Data obtained on a total of 16 treatments (combinations between moisture contents and exposure times to LN) were compared using one-way ANOVA and Tukey's test at $p \leq 0.05$. Percentage data were transformed for analysis according to y' = 2arcsine ((y/100)$^{0.5}$).

**3. Results**

At harvest, *T. labialis* seeds had a moisture content of 10.71 ± 0.14% FWB, while the viability was 95.55 ± 4.83%. Seed viability was not affected by different SMCs (Figure 2A). However, seed germination was affected when they reached a moisture content of 4% FWB (Figure 2B). In addition, the results showed an increase in the proportion of hard seeds when the moisture content decreased to 4% FWB (Figure 2C).

The SMC had no effect on seed germination after expansion at LN (Table 1). Although all treatments (after scarification in LN) significantly increased the percentage of germinated seeds (Figure 2B; Table 1), seeds exposed to LN for 15 min caused the lowest germination percentages, the highest percentages of hard seeds, and the lowest GI values. In addition, seed viability scores, MGT and T50, were improved in all SMCs studied when seeds were exposed to LN for at least 30 min. There were no statistical differences when seeds were exposed to LN for 30, 45, and 60 min. However, it was obvious that LN slightly affects seed viability by the number of dead seeds.

In Figure 3, the results show multiple cracks in seeds exposed to LN for 30, 45, and 60 min. Seeds not exposed to LN had a completely intact seed coat, whereas seeds exposed to LN for 15 min had small cracks in the macrosclereid cell layer.

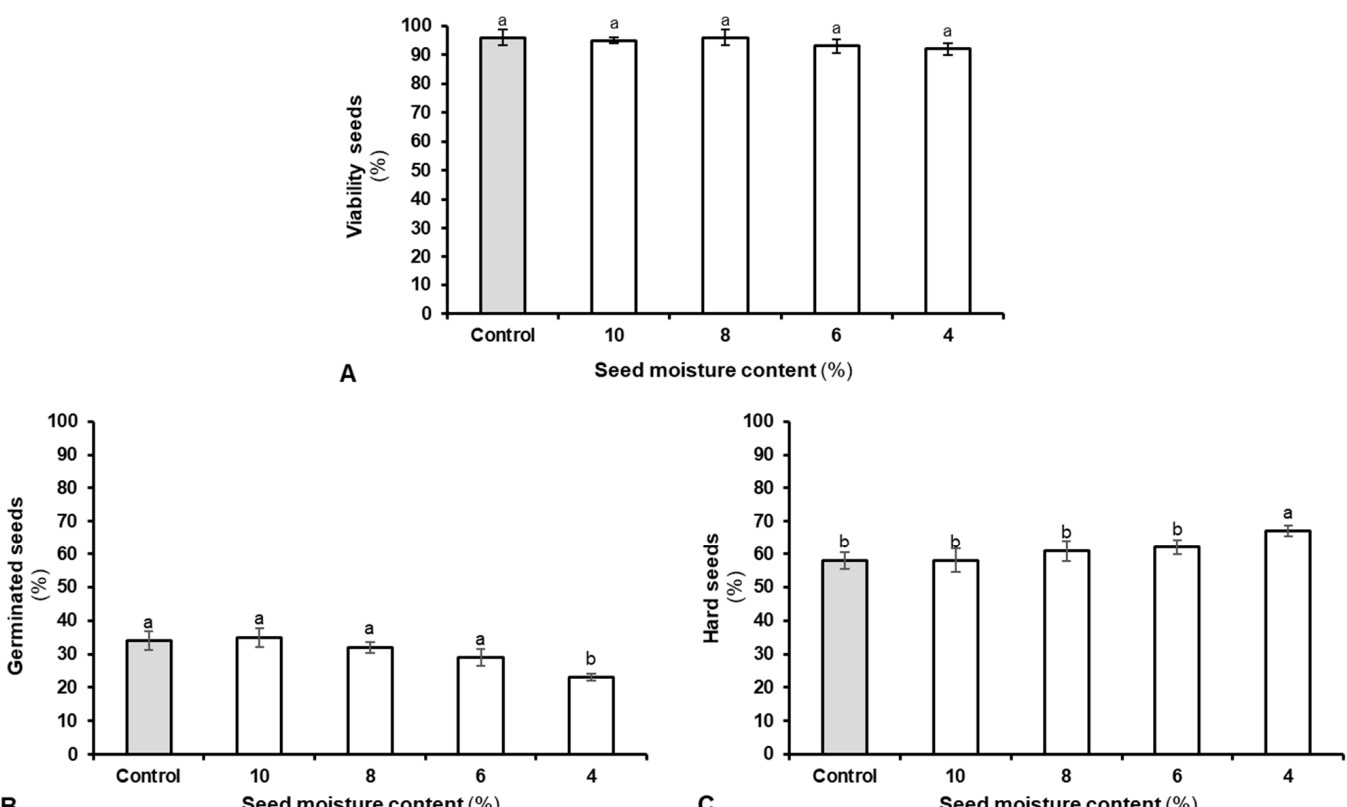

**Figure 2.** Quality of seeds of *T. labialis* (L.f.) Spreng after dehydration. (**A**) Viable, (**B**) germinated, and (**C**) hard seeds. Results with the same letter are not statistically different (one-way ANOVA; Tukey's test at $p \leq 0.05$). Percentage data were transformed for analysis according to y' = 2arcsine $((y/100)^{0.5})$. Vertical bars show the standard error.

**Table 1.** Germination after 28 days in a growth chamber of *T. labialis* seeds at different moisture contents and immersion times in liquid nitrogen (LN).

| Seed Moisture Content (%) | Immersion Time (min) | Germinated Seeds (%) | Hard Seeds (%) | Dead Seeds (%) | MGT (Days) | GI (Seeds Days$^{-1}$) | T$_{50}$ (Days) |
|---|---|---|---|---|---|---|---|
| 10 | 15 | 75 ± 1.9 (b) | 11 ± 1.01 (b) | 14 ± 0.98 (a) | 9.12 ± 0.56 (a) | 1.06 ± 0.09 (b) | 7.93 ± 0.48 (a) |
|  | 30 | 87 ± 1.63 (a) | 0 ± 0.00 (c) | 13 ± 1.02 (a) | 2.81 ± 0.21 (b) | 7.56 ± 0.63 (a) | 1.88 ± 0.09 (b) |
|  | 45 | 88 ± 2.51 (a) | 0 ± 0.00 (c) | 12 ± 1.03 (a) | 2.64 ± 0.19 (b) | 6.97 ± 0.61 (a) | 1.93 ± 0.18 (b) |
|  | 60 | 85 ± 2.58 (a) | 0 ± 0.00 (c) | 15 ± 1.11 (a) | 2.86 ± 0.25 (b) | 7.15 ± 0.58 (a) | 1.99 ± 0.12 (b) |
| 8 | 15 | 70 ± 2.46 (b) | 22 ± 1.93 (a) | 8 ± 0.31 (a) | 9.06 ± 0.45 (a) | 1.02 ± 0.08 (b) | 7.85 ± 0.66 (a) |
|  | 30 | 91 ± 1.91 (a) | 0 ± 0.00 (c) | 9 ± 0.68 (a) | 2.78 ± 0.16 (b) | 7.28 ± 0.45 (a) | 1.86 ± 0.15 (b) |
|  | 45 | 90 ± 1.92 (a) | 0 ± 0.00 (c) | 13 ± 1.01 (a) | 2.89 ± 0.22 (b) | 7.76 ± 0.65 (a) | 1.95 ± 0.12 (b) |
|  | 60 | 90 ± 1.23 (a) | 0 ± 0.00 (c) | 10 ± 0.58 (a) | 2.68 ± 0.21 (b) | 7.11 ± 0.57 (a) | 1.96 ± 0.11 (b) |
| 6 | 15 | 74 ± 2.26 (b) | 16 ± 1.35 (ab) | 14 ± 1.12 (a) | 9.05 ± 0.49 (a) | 1.12 ± 0.09 (b) | 7.44 ± 0.61 (a) |
|  | 30 | 89 ± 2.56 (a) | 0 ± 0.00 (c) | 11 ± 1.01 (a) | 2.61 ± 0.23 (b) | 7.31 ± 0.38 (a) | 1.94 ± 0.15 (b) |
|  | 45 | 87 ± 2.58 (a) | 0 ± 0.00 (c) | 13 ± 1.11 (a) | 2.74 ± 0.25 (b) | 7.05 ± 0.45 (a) | 1.98 ± 0.13 (b) |
|  | 60 | 89 ± 2.58 (a) | 0 ± 0.00 (c) | 11 ± 0.99 (a) | 2.63 ± 0.21 (b) | 7.28 ± 0.38 (a) | 1.89 ± 0.12 (b) |
| 4 | 15 | 67 ± 4.12 (b) | 22 ± 1.83 (a) | 11 ± 0.98 (a) | 8.86 ± 0.37 (a) | 0.95 ± 0.09 (b) | 7.45 ± 0.52 (a) |
|  | 30 | 88 ± 2.13 (a) | 0 ± 0.00 (c) | 12 ± 1.03 (a) | 2.58 ± 0.16 (b) | 6.98 ± 0.56 (a) | 1.89 ± 0.14 (b) |
|  | 45 | 89 ± 1.63 (a) | 0 ± 0.00 (c) | 11 ± 0.52 (a) | 2.63 ± 0.23 (b) | 7.01 ± 0.51 (a) | 1.86 ± 0.11 (b) |
|  | 60 | 86 ± 1.91 (a) | 0 ± 0.00 (c) | 14 ± 1.38 (a) | 2.55 ± 0.28 (b) | 7.26 ± 0.57 (a) | 1.81 ± 0.15 (b) |

(MGT) mean germination time, (GI) germination index, and (T$_{50}$) time for the germination of 50% of seeds. Results (mean ± SE determined (four replicates of 25 seeds, i.e., a total of 100 seeds per treatment) with different letters are statistically different per column (one-way ANOVA; Tukey's test at $p \leq 0.05$). Percentage data were transformed for analysis according to y' = 2arcsine $((y/100)^{0.5})$.

**Seeds not exposed to LN (control)**   **Seeds exposed to LN for 15 min**   **Seeds exposed to LN for 30 min**

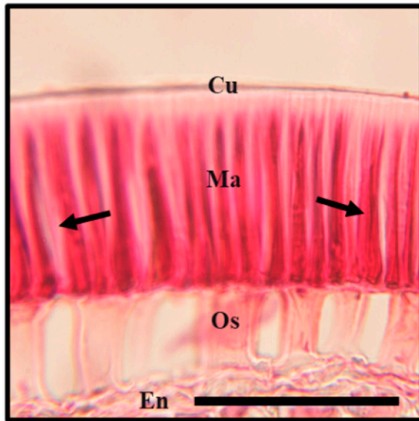
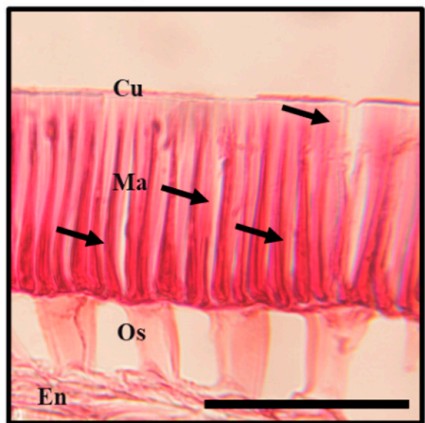

**Seeds exposed to LN for 45 min**   **Seeds exposed to LN for 60 min**

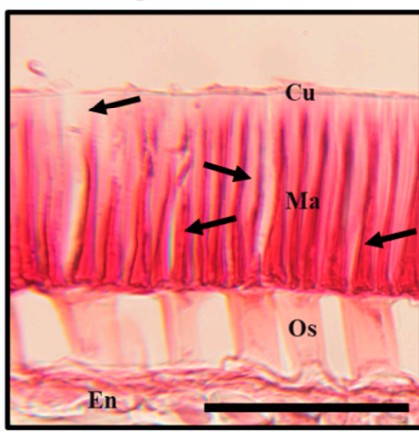
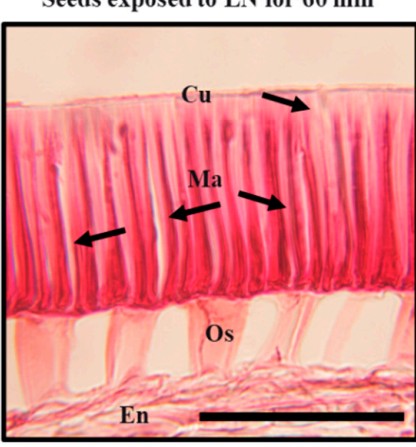

**Figure 3.** *Teramnus labialis* (L.f.) Spreng seed coat anatomy after different times of LN exposure: Cuticle (**Cu**); Endosperm remains (**En**); Macrosclereid cell layer (**Ma**); Osteosclereid cell layer (**Os**). Arrow represents crack. Bar = 50 μm.

## 4. Discussion

The dehydration of *T. labialis* seeds to a moisture content of 4% FWB did not affect initial viability. This water loss leads to severe stress conditions for the seeds [38,39]. According to the literature, the extreme desiccation of seeds may affect their initial viability if they are not of the orthodox type [40]. Therefore, *T. labialis* seeds may be considered orthodox in that they retain their viability after dehydration, as has been observed in *Glycine max* [41] and *Melanoxylon brauna* Schott [42] seeds.

The dehydration of *T. labialis* seeds to a moisture content of 4% FWB resulted in a decrease in the percentage of germination and an increase in the percentage of hard seeds. This could be due to the impermeability of the coat and seeds with PY, which increases with dehydration [43–45]. During dehydration, the cells of the seed coat shrink, resulting in mechanical compression and tighter packing. In addition, the natural entry ports for water are closed, which drastically reduces water uptake [44,46]. The results obtained for *T. labialis* are similar to those obtained for the seeds of different legume species [44,47–50].

The germination of *T. Labialis* seeds did not differ between moisture contents, although it is known that immersion in LN can be destructive to seeds, essentially due to the crystallization of water at the intracellular level [30,51,52]. This is because the moisture contents studied correspond to a hydration level and the water is strongly bound to hydrophilic and ionic sites. The latter contributes to the limitation of molecular mobility [53,54]. Therefore, the tissues are in a glassy state with no free water to form ice crystals [55]. The results obtained for *T. labialis* are in agreement with those of Cejas et al. [56], Naderi et al. [57], Endoh et al. [27], and Villalobos et al. [58] for seeds of other legume species, and with those of Zevallos et al. [59], Stegani et al. [60], Gobbi et al. [61], Villalobos et al. [62], and Vendrame et al. [63] for seeds of species of other families.

To eliminate the impermeability of the seed coat in the highest percentage of *T. labialis* seeds, expansion to LN was required from 30 to 60 min for all moisture contents. The thickness, chemical composition of the seed coat, size, and moisture content of the seeds, as well as the containers in which the seeds are placed, affect the rate at which the seeds cool and reach the cryogenic temperature [22,24,27,64–66].

The seeds of *T. labialis* were deposited in hermetically sealed polypropylene cryovials so that the LN could not penetrate inside. This factor determined the time required for the seeds to reach equilibrium with the cryogenic temperature and eliminate the impermeability of the seed coat. This result is in agreement with the results obtained when seeds were placed in polypropylene cryovials before immersion in LN [14,28,64,67,68].

The results obtained for the germination of *T. labialis* seeds correspond to the cracks in the seed coat observed at 30, 45, and 60 min of exposure. All indicators related to germination improved with the increase in cracks in the seed coat. Cracks in seed coats exposed to LN were first shown by Pritchard et al. [69] in seeds of *Trifolium arvense* L. In addition, Mira et al. [28] demonstrated the presence of cracks in the seed coat of *T. subterraneum* L., while Acosta et al. [9,10] showed cracks in the seed coat of *Teramnus labialis* as a result of seed immersion in LN. However, in this renewed study, cracks in the seed coat of *T. labialis* exposed to LN and germination showed a close relationship.

Data on the vigor of germinated seeds (MGT, GI, and $T_{50}$) correspond to the germination percentages. In this way, it is shown that the immersion time of seeds in LN determines the effectiveness of this scarification treatment in *T. labialis* seeds, in accordance with the breaking of dormancy. Our results are in agreement with the proposals of Mira et al. [28], Jastrzębowski et al. [64], and Endoh et al. [27].

## 5. Conclusions

The dehydration of *T. labialis* seeds at a moisture content of 4% FWB causes a decrease in germination capacity, but without a loss of viability. An increased moisture content does not affect the effectiveness of using LN as a scarification treatment. Only the duration of immersion in LN is a factor in determining the effectiveness of this treatment, because at least 30 min is required to dormancy break.

**Author Contributions:** Conceptualization, Y.A.F., L.P.G., M.E.M.M. and D.F.M.; investigation, methodology, and validation, Y.A.F., L.P.G. and D.F.M.; data curation and formal analysis, Y.A.F. and L.P.G.; writing—original draft preparation, Y.A.F.; writing—review, visualization and editing, Y.A.F. and M.E.M.M.; supervision and funding acquisition M.E.M.M. All authors have read and agreed to the published version of the manuscript.

**Funding:** This research received no external funding.

**Institutional Review Board Statement:** Not applicable.

**Informed Consent Statement:** Not applicable.

**Data Availability Statement:** Not applicable.

**Acknowledgments:** This research was supported by the Bioplantas Center and University of Ciego de Avila (Cuba).

**Conflicts of Interest:** The authors declare no conflict of interest.

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
