# Peer review of "Dormancy Breaking of Teramnus labialis (L.f.) Spreng Seeds Is Affected by the Extent of Liquid Nitrogen Exposure"

_2674-1024, doi:10.3390/seeds2010011_

Round 1

Reviewer 1 Report

The communication is interesting and presents a valuable collection of information on the use of liquid nitrogen as an effective scarification treatment to overcome physical dormancy of this plant species. In addition, the effect of seed dehydration on the viability and germination of T. labialis seeds, and the effect of moisture content and extent of LN exposure were also accessed.

I have highlighted a few points below that could improve the quality of the document before publication.

Wouldn't it be better to add the word "extent" instead of "time" in the title?

It is necessary to standardize the space between numbers and %; for example, 50% or 50%. Please correct this throughout the document following the journal's guidelines. Also check for double spaces throughout the manuscript; for example, lines 33, 35, 37, etc.

Consider adding an abbreviation for fresh weight basis, for example fresh weight basis (FWB); as this term is used several times throughout the document.

Line 15: “dormancy break” instead of “break dormancy”

Line 18: Moisture contents of xxx

Line 18: to LN for 15, xxx

Line 19: Seed viability did not change

Line 21: Seed moisture content did not xxxx

Line 22: was required to xxx

Line 23: increased from xx to xxx as the as the germination and the time of exposure to LN increased. “Please check that I have not changed the meaning of the sentence”

Line 26: Please use words that are not in the title

Line 37: Physical dormancy

Line 38: and air, been described as one of the main causes of xxxxx

Line 40: “agricultural development plants” Please add the reason for that.

Line 41: the seeds before sowing for successful germination.

Line 45: Describe “H2SO4”

Lines 48-49: to break dormancy and improve seed germination [9,10,21] and seedling establishment in various plant species [xx]

Lines 49-51: However, the efficiency of scarification treatment with LN is related to seed size, thickness and chemical composition of the seed coat [24], seed moisture content [25,26] and time of exposure to LN [27,28].

Line 55: during its immersion

Lines 56-59: In T. labialis species, using scanning electron microscopy, our group first demonstrated that seeds with a moisture content of 9.8% (fresh weight basis at harvest time) exposed to LN for 24 h had the formation of multiple cracks in the seed coat, resulting in xxxx [9].

Lines 59-61: Moreover, using light microscopy, we previously observed that seeds with a moisture content of 7.98% (fresh weight basis at harvest time) that were exposed to LN for 24 h had the hilar region opened and made the testa permeable to water, compromising its integrity [10].

Line 62: No need to have a new paragraph here. Consider moving this to line 61.

Line 63: effect of seed dehydration

Line 65: scarification an effective treatment.

Lines 69-70: The seed harvest was on March 12, 2018, when 80% of the pods were ripe.

Line 72: oven drying method “Describe the method please”

Line 72: The topographic test of 1 % tetrazolium “Describe the method please”

Line 75: were completely stained in dark red

Line 78: 4% of fresh xxx

Line 79: before scarifying them (400 seeds for each moisture content were used)

Line 86: Please add what was considered viability and germination

Line 92: and LN (-196°C) exposure

Line 93: for each moisture content were

Line 95: For warming, seeds were removed from cryovial and kept under room temperature (25°C) for 2 h [10].

Figure 1: Y axis: Use “Seed moisture content”

Figure 1: Use “min” instead of “minutes”

Lines 102-111: Long sentence…. consider rewriting the paragraph. this is confusing. For example: Seeds were scarified and controls (seed with x% FWB without any treatment) were placed on a single sheet of filter paper in 90 mm diameter plastic Petri-dishes (unsealed). The filter paper was moistened with 5 mL distilled water every 7 days. Four replicates of 25 seeds for each treatment were used for xxxx. Plates containing the seeds were moved into a growth chamber (TOP Cloudagri, RTOP-1000 B/D, China) set at constant temperature of 30 ± 1°C under a 16-h light/8-h dark photoperiod with a photosynthetic photon flux density of 80 μmol m−2 s −1 and 80% relative humidity. Four replicates of 25 seeds for each treatment were used and seeds were grown during 28 days [10]. The time for germination of 50% of the seeds (T50, days), the mean germinated time (MGT, days), and the germination index (GI, seeds days -1 ) were calculated (radicle of ≥ 2 mm was considered as germinated seed) [36].

Line 115: for histological and anatomical studies

Line 125: hard seeds? It's the first time you've mentioned it.

Line 126: and extend of LN exposure

Line 127: by two-way ANOVA and Tukey’s test at p ≤ 0.05.

Line 130: at harvest time

Lines 131-133: This long sentence can be replaced by: Seed viability was not affected by different seed moisture content Figure 2A).

Figure 2: X axis: Use “Seed moisture content”

Use Figure 2A; Figure 2B; Figure 2C instead of Figure 2, A; Figure 2, B; Figure 2, C

Line 140: One-Way ANOVA? Please check what you wrote in the material and methods

Line 140: by ?-way ANOVA and Tukey’s test at p ≤ 0.05.

Line 143-146: Long sentence…. consider rewriting the paragraph. this is confusing.

Line 148: times evaluated (Table 1).

Lines 149-151: move this paragraph after the table 1

Table 1: times to immersion? It's the first time you've mentioned it. Use a standard expression throughout the document

Describe the abbreviations in the footer of the table

Line 154-155: consider rewriting this sentence.. this is confusing. same to control seeds?

Lines 171-175: Long sentence…. consider splitting it into two sentences

Author Response

Dear reviewer

All your suggestions have been incorporated throughout the manuscript. We have tried to incorporate your valuable advice and suggestions wherever possible. We think that your comments have significantly improved the scientific value of the manuscript. With your comments:

P 1. General aspects:

Wouldn't it be better to add the word "extent" instead of "time" in the title?

It is necessary to standardize the space between numbers and %; for example, 50% or 50%. Please correct this throughout the document following the journal's guidelines. Also check for double spaces throughout the manuscript; for example, lines 33, 35, 37, etc.

Consider adding an abbreviation for fresh weight basis, for example fresh weight basis (FWB); as this term is used several times throughout the document.

A 1. All aspects have been considered in the new manuscript.

P 2. In the Abstract section:

Line 15: “dormancy break” instead of “break dormancy”

Line 18: Moisture contents of xxx

Line 18: to LN for 15, xxx

Line 19: Seed viability did not change

Line 21: Seed moisture content did not xxxx

Line 22: was required to xxx

Line 23: increased from xx to xxx as the as the germination and the time of exposure to LN increased. “Please check that I have not changed the meaning of the sentence”.

A 2. All aspects were corrected in the new manuscript.

P 3. In the Introduction section:

Line 37: Physical dormancy

Line 38: and air, been described as one of the main causes of xxxxx

Line 40: “agricultural development plants” Please add the reason for that.

Line 41: the seeds before sowing for successful germination.

Line 45: Describe “H2SO4”

Lines 48-49: to break dormancy and improve seed germination [9,10,21] and seedling establishment in various plant species [xx]

Lines 49-51: However, the efficiency of scarification treatment with LN is related to seed size, thickness and chemical composition of the seed coat [24], seed moisture content [25,26] and time of exposure to LN [27,28].

Line 55: during its immersion

Lines 56-59: In T. labialis species, using scanning electron microscopy, our group first demonstrated that seeds with a moisture content of 9.8% (fresh weight basis at harvest time) exposed to LN for 24 h had the formation of multiple cracks in the seed coat, resulting in xxxx [9].

Lines 59-61: Moreover, using light microscopy, we previously observed that seeds with a moisture content of 7.98% (fresh weight basis at harvest time) that were exposed to LN for 24 h had the hilar region opened and made the testa permeable to water, compromising its integrity [10].

Line 62: No need to have a new paragraph here. Consider moving this to line 61.

Line 63: effect of seed dehydration

Line 65: scarification an effective treatment.

A 3. All aspects were corrected in the new manuscript.

P 4. In the Materials and Methods section:

Lines 69-70: The seed harvest was on March 12, 2018, when 80% of the pods were ripe.

Line 72: oven drying method “Describe the method please”

Line 72: The topographic test of 1 % tetrazolium “Describe the method please”

Line 75: were completely stained in dark red

Line 78: 4% of fresh xxx

Line 79: before scarifying them (400 seeds for each moisture content were used)

Line 86: Please add what was considered viability and germination

Line 92: and LN (-196°C) exposure

Line 93: for each moisture content were

Line 95: For warming, seeds were removed from cryovial and kept under room temperature (25°C) for 2 h [10].

Figure 1: Y axis: Use “Seed moisture content”

Figure 1: Use “min” instead of “minutes”

Lines 102-111: Long sentence…. consider rewriting the paragraph. this is confusing. For example: Seeds were scarified and controls (seed with x% FWB without any treatment) were placed on a single sheet of filter paper in 90 mm diameter plastic Petri-dishes (unsealed). The filter paper was moistened with 5 mL distilled water every 7 days. Four replicates of 25 seeds for each treatment were used for xxxx. Plates containing the seeds were moved into a growth chamber (TOP Cloudagri, RTOP-1000 B/D, China) set at constant temperature of 30 ± 1°C under a 16-h light/8-h dark photoperiod with a photosynthetic photon flux density of 80 μmol m−2 s −1 and 80% relative humidity. Four replicates of 25 seeds for each treatment were used and seeds were grown during 28 days [10]. The time for germination of 50% of the seeds (T50, days), the mean germinated time (MGT, days), and the germination index (GI, seeds days -1 ) were calculated (radicle of ≥ 2 mm was considered as germinated seed) [36].

Line 115: for histological and anatomical studies

Line 125: hard seeds? It's the first time you've mentioned it.

Line 126: and extend of LN exposure

Line 127: by two-way ANOVA and Tukey’s test at p ≤ 0.05.

A 4. All aspects were taken into consideration and corrected in the new manuscript.

P 5. In the Results section:

Line 130: at harvest time

Lines 131-133: This long sentence can be replaced by: Seed viability was not affected by different seed moisture content Figure 2A).

Figure 2: X axis: Use “Seed moisture content”

Use Figure 2A; Figure 2B; Figure 2C instead of Figure 2, A; Figure 2, B; Figure 2, C

Line 140: One-Way ANOVA? Please check what you wrote in the material and methods

Line 140: by ?-way ANOVA and Tukey’s test at p ≤ 0.05.

Line 143-146: Long sentence…. consider rewriting the paragraph. this is confusing.

Line 148: times evaluated (Table 1).

Lines 149-151: move this paragraph after the table 1

Table 1: times to immersion? It's the first time you've mentioned it. Use a standard expression throughout the document.

Describe the abbreviations in the footer of the table.

A 5. Thanks to the reviewer. The authors corrected all aspects in the new document.

P 6. In the Discussion section:

Line 154-155: consider rewriting this sentence.. this is confusing. same to control seeds?

Lines 171-175: Long sentence…. consider splitting it into two sentences

A 6. The authors rewrite the sentences in the new manuscript.

With regards,

Reviewer 2 Report

The study sets out to investigate a role for liquid nitrogen treatment in the breaking of Teramnus labialis seed dormancy. According to the authors, liquid nitrogen treatments of at least 30 minutes are required to break Teramnus labialis physical seed dormancy, and this occurs through damage to the seed coat integrity, which changes the permeability of the seeds to increase germination. Overall, I have a few minor concerns that would need addressing:

1.     In lines 22, 25 and 209, for example (there are more throughout the text), the authors conclude that at least 30 min of liquid nitrogen exposure are necessary to break the dormancy of Teramnuss labialis seeds. With this, readers may infer that liquid nitrogen treatments shorter than 30 min will not break dormancy. However, in Table 1, seeds treated with liquid nitrogen for 15 min show a general increase of 4 % germination compared to the percentages shown in Figure 1B. To avoid ambiguity, I recommend that authors address these liquid nitrogen treatment differences explicitly.

2.     In Table 1, I suggest authors change “Death seeds” to “Dead seeds”. Were these seeds considered dead based on the tetrazolium assay? In terms of viability, Figure 1A shows that approximately 5-8% of the seeds are not viable (dead?). While in Table 1, after liquid nitrogen treatment, the numbers vary from 8–15 percent. Is this significant? If so, it is important to address in the text that liquid nitrogen treatments may also affect viability, though ever so slightly it may.

3. How did authors determine seed hardness (data shown in Figure 1C)?

4. In line 205, the authors conclude that dehydration of T. labialis seeds to a 4 percent moisture content decreased the permeability of the seed coat. Where is the data to support this conclusion?

Author Response

Dear reviewer #2,

Thank you for your time in revising our manuscript. All your suggestions have been incorporated throughout the manuscript. We have tried to incorporate your valuable advice and suggestions wherever possible. We think that your comments have significantly improved the scientific value of the manuscript.

We answered your following points:

P 1.     In lines 22, 25 and 209, for example (there are more throughout the text), the authors conclude that at least 30 min of liquid nitrogen exposure are necessary to break the dormancy of Teramnuss labialis seeds. With this, readers may infer that liquid nitrogen treatments shorter than 30 min will not break dormancy. However, in Table 1, seeds treated with liquid nitrogen for 15 min show a general increase of 4 % germination compared to the percentages shown in Figure 2B. To avoid ambiguity, I recommend that authors address these liquid nitrogen treatment differences explicitly.

A 1. In the new document we address this aspect in the LNs 156-162.

P 2.     In Table 1, I suggest authors change “Death seeds” to “Dead seeds”. Were these seeds considered dead based on the tetrazolium assay? In terms of viability, Figure 2A shows that approximately 5-8% of the seeds are not viable (dead?). While in Table 1, after liquid nitrogen treatment, the numbers vary from 8–15 percent. Is this significant? If so, it is important to address in the text that liquid nitrogen treatments may also affect viability, though ever so slightly it may.

A 2. We accomplished it in the new document.

P 3. How did authors determine seed hardness (data shown in Figure 2C)?

A 4. We mentioned in the LNs 118, 119 of the new document.

P 4. In line 205, the authors conclude that dehydration of T. labialis seeds to a 4 percent moisture content decreased the permeability of the seed coat. Where is the data to support this conclusion?

A 4. We corrected it in the new document.

With regards,

Reviewer 3 Report

The present article entitled “Dormancy Breaking of Teramnus labialis (L.f.) Spreng Seeds is Affected by Time of Liquid Nitrogen Exposition" mainly discussed the effect of dehydration on the viability and germination of T. labialis seeds and moisture content, adequate times of exposure to liquid nitrogen effectiveness treatment.

The experiment is well presented, and the results are significant for the agro-farming systems. The manuscript can be acceptable in the present form after improvement of language errors. Table 1, please revise the SE data values as 75±1.9b, 11±1.01a, 14±0.98a ……………. 

Author Response

Dear reviewer #3,

Thank you for your time in revising our manuscript. We have tried to improve the English language and reviewed the style and spelling throughout the document.

With regards

Reviewer 4 Report

Dear Authors,

Reading your paper was interesting. The practical contribution of article is great since the effect of moisture content of seeds and exposure time to liquid nitrogen on dormancy breaking and germination was investigated. The knowledge of the optimal treatment is necessary for the sexual propagation of this species.

However, all the comments and questions which were made on the manuscript (attached file) have to be taken account and answered.

Author Response

Dear reviewer #4,

Thank you for your time in revising our manuscript. All your suggestions have been incorporated throughout the manuscript. We have tried to incorporate your valuable advice and suggestions wherever possible. We think that your comments have significantly improved the scientific value of the manuscript.

We answered your following points:

P 1. In the previous document LNs 68 and 69 ¨march¨.

A 1. In the new document see LNs 67 and 68 ¨March¨.

P 2. In the previous document LN 79 you ask ¨How many seeds were there in each moisture content? Given that for each moisture content, 400 seeds were used for scarification treatments (100 for each of the 4 times) and seeds were also used to evaluate the viability and germination after dehydration¨.

A 2. In the new document see LNs 113-117 we specified ¨A total of 1250 seeds were selected. Three replicates of 50 seeds, i.e., a total of 150 seeds per treatment for each moisture content, including controls without dehydration, were evaluated for viability (%). At germination, four replicates of 25 seeds, i.e., a total of 100 seeds per treatment for each moisture content, were analyzed. Therefore, 750 seeds were used for viability (%) and 500 seeds for germination (%)¨. Moreover, we mentioned in the LNs 125-127 ¨For the 16th treatment combination, a total of 1600 seeds were used (with four replicates of 25 seeds, i.e., a total of 100 seeds per treatment for four moisture contents and 100 seeds for four immersion times in LN)¨.

P 3. In the previous document LN 85,86 you ask ¨Do the authors mean that a sample of 100 seeds from each moisture content was used for both tests (viability and germination)? The number of replicates and seeds per replicate for each test has to be referred¨.

A 3. In the new document we referred it in the LNs 81,92. Moreover, we specified our answer in point A 2.

P 4. In the original document, the reviewer asks regarding LN 93 ¨Why? Since 100 seeds from each of the four scarification durations (in total 400 seeds) were used for the germination test¨.

A 4. We corrected the new document as mentioned in points A 2 and A 3.

P 5. The reviewer mentioned that ¨The germinated seeds were  counted every how many days? This information has to be given in this section of the manuscript¨.

A 5. We corrected in LNs 97, 98 and also in the title of Table 1.

Please, more aspects that you mentioned are corrected in the new document.

With regards,

Round 2

Reviewer 1 Report

The authors have attended to the suggestions and covered all the points raised.